# Rolling Bearing Fault Diagnosis Based on WOA-VMD-MPE and MPSO-LSSVM

**DOI:** 10.3390/e24070927

**Published:** 2022-07-03

**Authors:** Zhihao Jin, Guangdong Chen, Zhengxin Yang

**Affiliations:** School of Mechanical and Power Engineering, Shenyang University of Chemical Technology, Shenyang 110142, China; jzh_sict_ln@sina.com (Z.J.); zhengxin1021@sina.com (Z.Y.)

**Keywords:** rolling bearing, whale algorithm, variational mode decomposition, Pearson correlation coefficient, K-fold cross-validation, multi-scale permutation entropy, modified particle swarm optimization, least square support vector machines

## Abstract

In order to further improve the accuracy of fault identification of rolling bearings, a fault diagnosis method based on the modified particle swarm optimization (MPSO) algorithm optimized least square support vector machine (LSSVM), combining parameter optimization variational mode decomposition (VMD) and multi-scale permutation entropy (MPE), was proposed. Firstly, to solve the problem of insufficient decomposition and mode mixing caused by the improper selection of mode component *K* and penalty factor *α* in VMD algorithm, the whale optimization algorithm (WOA) was used to optimize the penalty factor and mode component number in the VMD algorithm, and the optimal parameter combination (*K*, *α*) was obtained. Secondly, the optimal parameter combination (*K*, *α*) was used for the VMD of the rolling bearing vibration signal to obtain several intrinsic mode functions (IMFs). According to the Pearson correlation coefficient (PCC) criterion, the optimal IMF component was selected, and its optimal multi-scale permutation entropy was calculated to form the feature set. Finally, K-fold cross-validation was used to train the MPSO-LSSVM model, and the test set was input into the trained model for identification. The experimental results show that compared with PSO-SVM, LSSVM, and PSO-LSSVM, the MPSO-LSSVM fault diagnosis model has higher recognition accuracy. At the same time, compared with VMD-SE, VMD-MPE, and PSO-VMD-MPE, WOA-VMD-MPE can extract more accurate features.

## 1. Introduction

The rolling bearing is an important part of rotating machinery and equipment, whose main role is to transfer kinetic energy from the drive shaft to the shaft seat and reduce the energy loss caused by friction. A large part of the failure of rotating machinery and equipment is caused by rolling bearing failure. Rolling bearing failure will not only affect the progress of the project but also cause huge economic losses and, more seriously, will lead to staff casualties. Therefore, the study of rolling bearing fault diagnosis is necessary [1,2,3,4]. In the early days, staff mainly relied on manual experience to diagnose rolling bearings, and this method was inefficient and could not detect faults in the bearings at the earliest possible time. Later, it was found that the analysis of rolling bearing vibration signals could detect the status of bearings in real-time, so a large number of scholars studied various methods to process the signals. Dragomiretskiy [5] proposes variational mode decomposition (VMD), which is an adaptive signal decomposition method. Instead of adopting the same decomposition mode as empirical mode decomposition (EMD), this method adopts a non-recursive variational mode, which avoids the occurrence of the end effect and makes the decomposed mode components more accurate. However, the drawback of this method is that the number of mode components *K* and the penalty factor *α* have a large impact on the decomposition results [6]. To obtain the accurate number of mode components *K*, Zhou et al. [7] combined EMD and center frequency to determine the value of *K* according to the trend of center frequency variation of each intrinsic mode function (IMF). Zhang et al. [8] used the Gini index and autocorrelation function to construct the weighted autocorrelative function maximum (AFM) indicator as the optimization objective function and optimized the VMD using the improved particle swarm optimization (IPSO) algorithm to obtain the required parameters *K* and *α* for the VMD decomposition to obtain the sensitive IMFs. Wang et al. [9] used the Archimedes optimization algorithm (AOA) to optimize the mode number *K* and penalty factor *α* of the VMD algorithm by taking the minimum average value of all IMFs’ correlation waveform index (Cwi) as the objective function. Jiao et al. [10] determined the mode number *K* required for VMD decomposition according to the method of abnormal decline of center frequency (ADCF). Duan et al. [11] combined the improved VMD and sample entropy (SE) to determine the value of *K* by the maximum correntropy criterion (MCC), which effectively improved the statistical properties of highly nonlinear process errors. Li et al. [12] proposed a genetic algorithm (GA) to optimize VMD decomposition parameters *K* and *α*, which decomposes the optimal IMFs and improves the accuracy of VMD decomposition. Extracting appropriate feature information is the key that determines the accuracy and reliability of fault diagnosis results. He et al. [13] used an improved sparrow search algorithm to optimize the VMD parameters with dispersion entropy as the fitness value and used the optimized VMD algorithm to decompose the original signal into a series of mode components and calculate the energy entropy of each mode component to complete the flywheel bearing fault diagnosis. Xue et al. [14] calculated the dispersion entropy of IMF components in different frequency bands and then used the joint approximate diagonalization of eigenmatrices (JADE) to extract fusion features and finally obtain the hierarchical discrete entropy (HDE) for bearing fault diagnosis. Wang et al. [15] proposed a feature extraction method based on the combination of variational mode extraction (VME) and multi-objective information fusion band-pass filter (MIFBF). Yang et al. [16] used the fractional Fourier transform (FRFT) algorithm to extract fault features from the original signals and then used stochastic resonance (SR) to enhance the weak fault feature information to complete bearing fault diagnosis according to the fault feature frequency. Yan et al. [17] performed VMD decomposition of bearing signals, and the calculated multi-scale envelope dispersion entropy (MEDE) of the IMF component was used as the feature to complete bearing fault pattern recognition. Zheng et al. [18] calculated the permutation entropy (PE) value of each IMF obtained by VMD decomposition to reflect the characteristic information of the bearing vibration signal. Zhang et al. [19] combined VMD and sample entropy and used the multi-domain indexes to construct the feature vector to characterize the fault information.

An intelligent fault diagnosis method is needed for pattern recognition of rolling bearings in order to enable rapid fault diagnosis of fault characteristic information and avoid mechanical equipment failures. Vapnik [20] proposed the support vector machine (SVM) machine learning algorithm mainly to solve the problems of nonlinearity as well as insufficient samples. Zhang et al. [21] used multi-scale information entropy to construct a sample set, and IPSO optimization SVM was used to realize bearing fault diagnosis. Wang et al. [22] used quantum-behaved particle swarm optimization (QPSO) and multi-scale permutation entropy (MPE) to extract features from denoising bearing signals and then used SVM to identify faults. The experimental results show that the proposed fault diagnosis method can identify bearing fault types well. Ye et al. [23] used VMD-MPE to construct feature vectors, then used PSO to optimize SVM to improve the model recognition accuracy. However, SVM is complicated to solve the non-equation constraint problem, and in order to reduce the solution difficulty, Suykens [24] improved SVM and proposed the least square support vector machine (LSSVM), which replaced the non-equation constraint in SVM with an equation constraint, greatly reducing the solution difficulty. The LSSVM algorithm has been widely applied in the field of industrial intelligence in recent years [25,26,27,28]. He et al. [29] used wavelet packet transform to extract fault features and combined them with LSSVM to complete the fault identification of circuit output voltage signals. Gao et al. [30] fused singular entropy, energy entropy, and permutation entropy to obtain complementary features, combined with the PSO algorithm to optimize LSSVM, and successfully completed the diagnosis of bearing faults. Zhao et al. [31] extracted narrowband kurtosis vectors from the cyclic correntropy spectrum (CCES) as feature vectors of LSSVM for the early detection and classification of locomotive axle bearing faults. Zhu et al. [32] used VMD to decompose the bearing vibration signal, used the fuzzy entropy of each IMF as the feature vector, optimized the LSSVM model by the gray wolf optimizer (GWO) algorithm, and finally completed the identification of the rolling bearing faults.

The methods in the above literature simply perform individual optimization of feature extraction or model parameters, which limits the accuracy of rolling bearing fault diagnosis. The future trend is definitely to optimize feature extraction and model parameters simultaneously with different algorithms to avoid the problem of low accuracy caused by individual optimization. In this paper, the whale algorithm (WOA) is used to optimize the VMD algorithm, and the optimal combination of parameters (*K*, *α*) required for VMD decomposition is obtained. According to the Pearson correlation coefficient (PCC) criterion, the optimal IMF component is selected, and its optimal multi-scale permutation entropy is calculated to form the feature set. Finally, k-fold cross-validation was used to train the MPSO-LSSVM model, and the test set was input into the trained model for identification. The experimental results show that compared with PSO-SVM, LSSVM, and PSO-LSSVM, the MPSO-LSSVM fault diagnosis model has higher recognition accuracy. Meanwhile, compared with VMD-SE, VMD-MPE, and PSO-VMD-MPE, WOA-VMD-MPE can extract more accurate features.

## 2. Feature Extraction

The first step of establishing a rolling bearing diagnosis model is feature extraction. Whether the extracted features are accurate or not directly determines the accuracy of diagnosis, so the extracted features must be able to truly and accurately reflect the status information of the bearing. Since different parts produce different frequencies of vibration signals, this will lead to different IMFs after VMD decomposition, and the calculated multi-scale permutation entropy values of IMFs will be different according to which feature information will be constructed. In feature extraction, a series of IMFs are obtained by WOA-VMD decomposition of the vibration signal, and the multi-scale permutation entropy value of each IMF is calculated as the feature vector.

### 2.1. VMD

VMD is an adaptive signal decomposition method that uses a non-recursive decomposition mode to decompose the signal into a specified number of IMFs with different center frequencies according to a predetermined number of modes *K* and a penalty factor *α*. It gets rid of the uncertainty of the number of IMFs caused by the traditional method of EMD decomposition as well as the end effect and modal mixing problems encountered and can better highlight the characteristic information of the signal [33]. The expression of the *k*-th order eigenmode function is obtained by VMD decomposition, that is:(1)$$u_{k}(t)\;=\;A_{k}(t)\cos\left(\phi_{k}(t)\right)$$
(2)$$\omega_{k}(t)\;=\;\phi_{k}^{\prime }(t)\;=\;\frac{d\phi_{k}(t)}{dt}$$
where $A_{k}(t)$ is the instantaneous amplitude of $u_{k}(t)$, k = (1,2,…,K). $\omega_{k}(t)$ is the center frequency of $u_{k}(t)$. $\phi_{k}(t)$ is a non-monotonically decreasing phase function.

The analytical signal of $u_{k}(t)$ is obtained by the Hilbert transform, so as to obtain the unilateral frequency spectrum, that is:(3)$$\left[\delta (t)\;+\;\frac{j}{\pi t}\right]\;*\;u_{k}(t)$$

By adjusting the center frequency $\omega_{k}(t)$ of each $u_{k}(t)$ and mixing it with the unilateral frequency spectrum of each mode, the baseband signal is obtained:(4)$$\left[\left(\delta (t)\;+\;\frac{j}{\pi t}\right)\;*\;u_{k}(t)\right]e^{-j\omega_{k}t}$$

Calculate the square of the $L^{2}$ norm of the gradient of the demodulated signal to obtain the bandwidth of the demodulated signal, and establish the following constrained variational model expression:(5)$$\left\{\begin{array}{l} \underset{\left\{u_{k}\right\}\left\{\omega_{k}\right\}}{\min}\left\{{\sum_{k=1}^{K}\left\|\partial t\left[\left(\delta (t)\;+\;\frac{j}{\pi t}\right)\;*\;u_{k}(t)\right]e^{-j\omega_{k}t}\right\|}_{2}^{2}\right\}\\ s.t.\sum_{k=1}^{K}u_{k}(t)\;=\;f(t) \end{array}\right.$$
where f(t) is the input signal, and δ(t) is the pulse function.

In order to turn Equation (5) into an unconstrained variational problem and to ensure the accuracy of the signal decomposition, an extended Lagrange function is introduced, whose expression is:(6)$$L\left(\left\{u_{k}\right\},\left\{\omega_{k}\right\},\lambda \right)\;=\;\alpha {\sum_{k=1}^{K}\left\|\partial t\left[\left(\delta (t)\;+\;\frac{j}{\pi t}\right)\;*\;u_{k}(t)\right]e^{-j\omega_{k}t}\right\|}_{2}^{2}\;+\;{\left\|f(t)\;-\;\sum_{k=1}^{K}u_{k}(t)\right\|}_{2}^{2}\;+\;\langle \lambda (t),f(t)\;-\;\sum_{k=1}^{K}u_{k}(t)\rangle$$
where α is the quadratic penalty factor, λ is the Lagrange operator, and 〈,〉 represents the inner product.

Use the alternate direction method of multipliers (ADMM) to continuously update ${\overset{\wedge }{u}}_{k}^{n+1}(\omega )$, $\omega_{k}^{n+1}$, and ${\overset{\wedge }{\lambda }}^{n+1}(\omega )$ alternately to find the minimum value of Equation (6).
(7)$${\overset{\wedge }{u}}_{k}^{n+1}(\omega )\;=\;\frac{f(\omega )\;-\;\sum_{i<k}{\overset{\wedge }{u}}_{i}^{n+1}(\omega )\;-\;\sum_{i>k}{\overset{\wedge }{u}}_{i}^{n}(\omega )\;+\;{\overset{\wedge }{\lambda }}^{n}(\omega )/2}{1\;+\;2\alpha {\left(\omega \;-\;\omega_{k}^{n}\right)}^{2}}$$
(8)$$\omega_{k}^{n+1}\;=\;\frac{\int_{0}^{\infty }\omega {\left|{\overset{\wedge }{u}}_{k}^{n+1}(\omega )\right|}^{2}d\omega }{\int_{0}^{\infty }{\left|{\overset{\wedge }{u}}_{k}^{n+1}(\omega )\right|}^{2}d\omega }$$
(9)$${\overset{\wedge }{\lambda }}^{n+1}(\omega )\;=\;{\overset{\wedge }{\lambda }}^{n}(\omega )\;+\;\tau \left(f(\omega )\;-\;\sum_{k=1}^{K}{\overset{\wedge }{u}}_{k}^{n+1}(\omega )\right)$$

The iteration ends when the accuracy satisfies Equation (10) and, finally, *K* IMFs are obtained.
(10)$$\sum_{k=1}^{K}\frac{{\left\|{\overset{\wedge }{u}}_{k}^{n+1}\;-\;{\overset{\wedge }{u}}_{k}^{n}\right\|}_{2}^{2}}{{\left\|{\overset{\wedge }{u}}_{k}^{n}\right\|}_{2}^{2}}\;<\;\varepsilon$$
where *ε* (*ε* > 0) is the precision convergence value.

According to the above theoretical analysis, the specific process of the VMD algorithm is as follows:
Step 1. Initialize $\left\{u_{k}^{1}(\omega )\right\}$,$\left\{\omega_{k}^{1}\right\}$,$\lambda_{k}^{1}(\omega )$ and *n* = 0.Step 2. Let *n* = *n* + 1, the loop starts.for *k* = 1:*K*update ${\overset{\wedge }{u}}_{k}^{n+1}(\omega )$, $\omega_{k}^{n+1}$ and ${\overset{\wedge }{\lambda }}^{n+1}(\omega )$Step 3. Given the precision, if the iteration stop condition is met, stop the loop; otherwise, enter step 2 and continue the loop.

### 2.2. PCC

A Pearson correlation coefficient (PCC) is used to measure the linear correlation between two sets of data, that is, to carry out a correlation analysis between variables and select variables with strong correlation [34]. The closer the absolute value is to 1, the stronger the correlation between variables. The formula for calculating is as follows:(11)$$\rho_{X,Y}\;=\;\frac{\mathrm{cov}(X,Y)}{\sigma_{X}\sigma_{Y}}\;=\;\frac{E((X\;-\;\mu_{X})(Y\;-\;\mu_{Y}))}{\sigma_{X}\sigma_{Y}}\;=\;\frac{E(XY)\;-\;E(X)E(Y)}{\sqrt{E(X^{2})\;-\;E^{2}(X)}\sqrt{E(Y^{2})\;-\;E^{2}(Y)}}$$
where *E* is the mathematical expectation, cov is the covariance, and *σ* is the standard deviation.

According to the literature [35], it can be known that the signal component with a correlation number greater than 0.3 should be selected. This eliminates irrelevant features and avoids losing sensitive fault signal information.

### 2.3. MPE

Permutation entropy (PE) can detect the complexity and randomness of time series and is sensitive to local variations, so it is usually used for mechanical equipment fault diagnosis [36]. However, PE can only reflect the complexity of time series at a single scale and cannot reflect the situation at multiple scales, so the MPE is introduced. The MPE is used to determine the complexity and randomness of a time series by calculating the PE of the time series at multiple scales [37]. The calculation procedure is as follows:

Given a time series $X\;=\;\{x_{i},i\;=\;1,2,\ldots ,N\}$ of length *N*, a time series $y_{j}^{\left(s\right)}$ with a scale factor *s* is obtained by coarse granulation:(12)$$y_{j}^{\left(s\right)}\;=\;\frac{1}{s}\sum_{i=(j-1)s+1}^{js}x_{i},\left(j\;=\;1,2,\ldots ,\frac{N}{s}\right)$$

Reconstructing the time series $y_{j}^{\left(s\right)}$ according to the embedding dimension *m* and the delay time *t* yields $Y_{l}^{\left(s\right)}$:(13)$$Y_{l}^{\left(s\right)}\;=\;\left\{y_{l}^{\left(s\right)},y_{l+t}^{\left(s\right)},\ldots ,y_{l+(m-1)t}^{\left(s\right)}\right\}$$

The reconstructed components of Equation (12) are arranged in increasing order to obtain the sign vector, which is:(14)$$S(k)\;=\;(j_{1},j_{2},\ldots ,j_{m}),k\;=\;1,2,\ldots ,K(K\;\leq \;m!)$$


According to the probability of occurrence of each sign, the MPE can be defined as:(15)$$H_{p}\;=\;-\sum_{k=1}^{K}P_{k}\ln P_{k}$$

The smaller the value of $H_{p}$, the more orderly the time series is and the more likely it is to be in a fault state; the larger the value of $H_{p}$, the more irregular the time series is and the greater the probability that it is in a normal state.

### 2.4. Feature Extraction Based on WOA-VMD and MPE

Mirjalili [38] proposed a novel population intelligence optimization algorithm, the whale optimization algorithm (WOA), based on the hunting behavior of humpback whales. This algorithm can effectively avoid falling into the trap of local minima, and the global optimization search is more effective. Since the mode number *K* and penalty factor *α* in the VMD algorithm have a large impact on the decomposition results, this paper uses WOA to optimize the parameters *K* and *α*.

The WOA needs to define a suitable fitness function to calculate the fitness value when optimizing the VMD parameters and update the parameters by comparing the fitness values. In this paper, the envelope entropy is chosen as the fitness function. The size of the envelope entropy value reflects the uncertainty of the probability distribution, and the larger the entropy value is, the more uncertain the signal is. The envelope entropy $E_{p}$ of the signal x(t)(t = 1,2,…,N) is calculated as follows:(16)$$\left\{\begin{array}{l} E_{p}\;=\;-\sum_{t=1}^{N}p_{t}\mathrm{lg}p_{t}\\ p_{t}\;=\;\frac{a(t)}{\sum_{t=1}^{N}a(t)} \end{array}\right.$$
where *N* is the number of signal sampling points, a(t) is the envelope signal obtained by Hilbert demodulation of signal x(t), and $p_{t}$ is the normalization result of signal a(t).

The flow chart of feature extraction based on WOA-VMD and MPE is shown in Figure 1. The specific steps are as follows:
Step 1. Initialize the WOA algorithm, take the envelope entropy as the fitness function of the WOA, and obtain the global optimal parameters (*K*, *α*) for the VMD decomposition of the signal.Step 2. VMD decomposition of the vibration signal according to (*K*, *α*) obtained in step 1 to obtain *K* IMF components and pick the best ones according to the PCC criterion.Step 3. Select the optimal MPE parameters and calculate the MPE value of each IMF to form the feature data set.

## 3. Establishment of Fault Diagnosis Model

### 3.1. MPSO

The particle swarm optimization (PSO) algorithm is a global optimization algorithm with an efficient search function. However, it is easy to fall into the local optimum, the accuracy decreases in the late iteration, and the convergence speed is slow when searching for the best [39], so this paper proposes the improved particle swarm optimization (MPSO) algorithm. MPSO adopts linear decreasing weights and time-varying learning factors to optimize PSO, which improves the search ability and convergence speed of the algorithm. In the MPSO optimization principle, in a *D*-dimensional vector, the position of the *p*-th particle is $X_{p}\;=\;(x_{p1},x_{p2},\ldots ,x_{pD})$, the velocity is $v_{p}\;=\;(v_{p1},v_{p2},\ldots ,v_{pD})$, the optimal position of the particle is $W_{p}\;=\;(p_{p1},p_{p2},\ldots ,p_{pD})$, and the optimal position of all particles is $W_{g}\;=\;(w_{g1},w_{g2},\ldots ,w_{gD})$.

The velocity and position update equations are as follows:(17)$$v_{p}^{k+1}\;=\;\omega v_{p}^{k}\;+\;c_{1}r_{1}(W_{p}\;-\;X_{p}^{k})\;+\;c_{2}r_{2}(W_{g}\;-\;X_{p}^{k})$$
(18)$$X_{p}^{k+1}\;=\;X_{p}^{k}\;+\;v_{p}^{k+1}$$
where ω is the inertia weight; $c_{1}$ and $c_{2}$ are the learning factor constants; and $r_{1}$ and $r_{2}$ are uniform random numbers in the range of [0, 1].

The inertia weight ω represents the ability of the particle to maintain the velocity of motion at the previous moment. When the value of ω is small, the local search ability is stronger, and when the value of ω is larger, the global search ability is stronger. In the early stages of the search, the global search ability needs to be improved to avoid getting into local optimal solutions, and in the later stages of the search, the local search ability needs to be improved to find optimal solutions. The linear decreasing inertia weights can better balance the global and local search ability of the algorithm, and the expression is as follows:(19)$$\omega \;=\;\omega_{\max}\;-\;\frac{g(\omega_{\max}\;-\;\omega_{\min})}{g_{\max}}$$
where $\omega_{\max}$ is the maximum value of inertia weight, $\omega_{\min}$ is the minimum value of inertia weight, g is the current number of iterations, and $g_{\max}$ is the maximum number of iterations.

The learning factor $c_{1}$ represents particle self-awareness and $c_{2}$ represents particle social awareness. In order to facilitate particle search, it is necessary to improve self-awareness in the early stages of the search and social awareness in the latter stages. The expression of the learning factor is:(20)$$\left\{\begin{array}{l} c_{1}\;=\;(c_{1f}\;-\;c_{1s})\frac{g}{g_{\max}}\;+\;c_{1s}\\ c_{2}\;=\;(c_{2f}\;-\;c_{2s})\frac{g}{g_{\max}}\;+\;c_{2s} \end{array}\right.$$
where $c_{1s}$ and $c_{1f}$ are the initial and final values of $c_{1}$; $c_{2s}$ and $c_{2f}$ are the initial and final values of $c_{2}$ and are constants.

### 3.2. LSSVM Fault Diagnosis Model Based on MPSO Optimization

The selection of the regularization parameter *γ* and the radial basis kernel function parameter *σ* in the LSSVM model with radial basis function (RBF) as the kernel function is critical when classifying faults in rolling bearings, and the improper selection of the parameters will lead to poor classification model results. The initial value selection in the pre-classification stage is random, and in the past, it relied on experience to select the appropriate parameters, which can cause the problem of underfitting or overfitting to occur. The MPSO algorithm is used to optimize the parameter combination (*γ*, *σ*), which avoids the above disadvantages and greatly improves the classification accuracy of the LSSVM model. The specific process is shown in Figure 2.

The MPSO-LSSVM steps are as follows:
Step 1: Extract the fault features of rolling bearing vibration signal processing and construct them into a training set and a test set.Step 2: Initialize particle swarm parameters. The dimension is two because the parameter combination (*γ*, *σ*) is optimized. The parameters of the algorithm are set and the initial swarm of particles is generated randomly.Step 3: Calculate the accuracy error $\delta_{e}$ of each particle as the fitness value through Equation (20), and the smaller the fitness value, the better the diagnosis result of the LSSVM model. That is:(21)$$\delta_{e}\;=\;1\;-\;\frac{r_{x}}{r_{x}\;+\;r_{y}}$$
where $r_{x}$ is the number of correct classifications and $r_{y}$ is the number of wrong classifications.Step 4: According to the particle fitness, the velocity and position of the particle are updated by Equations (16) and (17).Step 5: If the maximum number of iterations or the termination condition is satisfied, the loop ends and the optimal combination of parameters is output to construct the MPSO-LSSVM model. Otherwise, return to Step 4.Step 6: Input the test set into the constructed MPSO-LSSVM model to obtain the fault diagnosis result.

## 4. Experiment

This paper adopts the Western Reserve University bearing test bench data to verify the method [40]. Figure 3 shows a diagram of the experimental setup. The main shaft of the motor is supported by the fan end (FE) and drive end (DE) bearings, respectively, and the bearings are pitted by EDM to simulate common failures. The vibration signals of the drive-side bearing type SKF 6205 2RS acquired by a 16-way DAT recorder with a motor speed of 1797 r/min, a sampling frequency of 12 kHz, and a load of 0 hp are used in the experiments. The vibration signals are collected in four states: the normal (Normal), the inner race fault (IRF), the outer race fault (ORF), and the ball fault (BF). In this study, 100 groups of each state are sampled, and 400 groups of four states are sampled, with 1024 sampling points per group.

Using the signal of the bearing inner race fault as an example, the WOA algorithm is used to find the optimal parameter combination (*K*, *α*) of VMD decomposition. To verify the effectiveness of WOA in VMD parameter optimization, PSO-VMD and GA-VMD are used to compare and verify WOA-VMD, respectively. The initial parameters are as follows: the maximum iteration number is 40, the population size is 20, the average value of 20 tests is taken, the range of *K* is [2, 10], and the range of *α* is [500, 6000]. The convergence comparison of the three optimization algorithms is shown in Figure 4.

It can be seen from Figure 4 that PSO-VMD, GA-VMD, and WOA-VMD converge at the 12th, 18th, and 26th generations, respectively, and the convergence value is 3.4045. The convergence speed of the WOA-VMD fitness value optimization curve is the fastest. Table 1 shows the average time taken to run VMD under three optimization algorithms.

It can be seen from Table 1 that GA-VMD runs the longest and WOA-VMD runs the shortest. It shows that the WOA-VMD algorithm has advantages over the GA-VMD and PSO-VMD algorithms.

The WOA-VMD optimization algorithm is used to optimize the four bearing signals, and the fitness curve is shown in Figure 5. Figure 5a shows that after 14 iterations, the best fitness of the normal signal is obtained, the convergence value is 3.2228, and the best parameter combination (*K*, *α*) is (9, 2103). Figure 5b shows that after 12 iterations, the best fitness of the inner race fault signal is obtained, the convergence value is 3.4045, and the best parameter combination (*K*, *α*) is (6, 3648). Figure 5c shows that after 15 iterations, the best fitness of the outer race fault signal is obtained, the convergence value is 3.2066, and the best parameter combination (*K*, *α*) is (7, 2585). Figure 5d shows that after nine iterations, the best fitness of the ball fault signal is obtained, the convergence value is 3.0738, and the best parameter combination (*K*, *α*) is (9, 3029). The result of the four data optimizations is shown in Table 2.

The bearing signals are VMD decomposed, and the decomposition results are shown in Figure 6, Figure 7, Figure 8 and Figure 9. Figure 6a, Figure 7a, Figure 8a and Figure 9a show the time-domain waveform. The frequency-domain analysis is performed on the decomposed IMF, and its frequency spectrum is shown in Figure 6b, Figure 7b, Figure 8b and Figure 9b. It can be seen from Figure 6b, Figure 7b, Figure 8b and Figure 9b that the IMFs have different central frequencies and no defects such as state aliasing and signal distortion, and the original signal can be effectively decomposed.

According to the PCC, the Pearson correlation coefficients between each IMF and the original signal are calculated. The calculated results are shown in Table 3, Table 4, Table 5 and Table 6.

It can be seen from Table 3 that the components of IMF1, IMF2, IMF3, and IMF5 obtained by VMD decomposition of the normal (Normal) signal meet the PCC condition with a correlation value greater than 0.3. This indicates that IMF1, IMF2, IMF3, and IMF5 components are highly correlated with the original signal, and the signal contains abundant fault information. Therefore, IMF1, IMF2, IMF3, and IMF5 components are selected as the key components. It can be seen from Table 4 that the IMF3, IMF4, IMF5, and IMF6 components obtained by VMD decomposition of the outer race fault (ORF) signal meet the PCC condition with correlation values greater than 0.3. Therefore, IMF3, IMF4, IMF5, and IMF6 components are selected as the key components. It can be seen from Table 5 that the IMF4, IMF5, IMF6, and IMF7 components obtained by VMD decomposition of the ball fault (BF) signal meet the PCC condition with a correlation value greater than 0.3. Therefore, IMF4, IMF5, IMF6, and IMF7 components are selected as the key components. It can be seen from Table 6 that the components of IMF2, IMF3, IMF4, IMF5, and IMF6 obtained by VMD decomposition of the inner race fault (IRF) signal meet the PCC condition with a correlation value greater than 0.3. According to the above analysis, the normal (Normal) signal, the outer race fault (ORF) signal, and the ball fault (BF) signal only have four IMF components that satisfy the PCC condition. Therefore, in order to ensure the same dimension of the eigenvectors obtained below, the IMF3, IMF4, IMF5, and IMF6 components are selected as optimal components. A new array can be formed based on the order of the optimal IMF components obtained above. The result is shown in Table 7.

The selection of MPE parameters is extremely important and determines the accuracy of fault diagnosis. The method of determining the optimal MPE parameters is introduced, initially setting the embedding dimension *s* = 6, the delay time *t* = 1, and setting the scale factor to *τ* = 20. Figure 10 shows the relationship between MPE values of the array U and scale factor *τ*. It can be seen from Figure 10a that when *τ* = 2, the difference in MPE value is larger, and four states can be clearly distinguished. Therefore, the value of the optimal scale factor for U1 is determined as 2 and uses the same method to determine the optimal scale factor *τ* = 4, *τ* = 9, and *τ* = 5 of U2, U3, and U4. The result is shown in Table 8.

According to the optimal scale factor *τ*, the optimal MPE is selected to form the feature vectors. The feature vectors in the four states are normalized to the range of (0, 1) to form the feature vector data set, for which Table 9 shows the feature vector data set. Figure 11 is the boxplot of the feature vector U for the four types of bearing signals. It can be seen from Figure 11 that the feature vectors are relatively concentrated.

## 5. Analysis of Fault Diagnosis Results

The feature vector data set is input into the LSSVM model for classification, and the MPSO algorithm is used to optimize the model. The parameters of the MPSO-LSSVM algorithm are set as follows: the $c_{1s}$, $c_{1f}$, $c_{2s}$, and $c_{2f}$ values are 2, 1, 1, 2, the $\omega_{\max}$ is 0.9, the $\omega_{\min}$ is 0.1, the number of particles is 30, the number of iterations is 200, the penalty factor range is [0.1, 100], and the radial basis kernel parameter range is [0.1, 100]. For the multi-classification problem, the sample data is grouped and trained by the K-fold cross-validation method. *K* = 10 is selected, each subset of data is used as a validation set, and the remaining nine sets of subset data are combined as a training set, which is brought into the MPSO-LSSVM model for training. The accuracy rate of 10 groups of discriminant models obtained through training is shown in Figure 12, and the average accuracy is 99.75%, which proves that the model can perfectly discriminate the fault types of rolling bearings and effectively avoid the effects of over-fitting.

In order to verify whether the model trained by K-fold cross-validation has excellent generalization ability, a total of 80 test samples are classified and identified by taking 20 test samples of the normal, inner race fault, outer race fault, and ball fault. The fitness curve of the algorithm with the number of iterations is shown in Figure 13. The MPSO algorithm optimizes the optimal combination of LSSVM parameters (*γ*, *σ*) as (30.65, 7.13), and the accuracy of the model is 99.88%. The classification result is shown in Figure 14. It can be seen from Figure 14 that the classification rate is 100%.

In order to prevent the contingency of experimental results, the test set is tested 20 times and takes an average of 20 results. Table 10 shows the diagnosis results. As can be seen from Table 10, the average accuracy is 100% after optimizing the LSSVM model with the MPSO algorithm, which proves that the MPSO-LSSVM pattern recognition has a strong adaptive capability.

To further verify the superiority of this model, the same bearing faults are diagnosed by combining the feature vectors constructed by WOA-VMD-MPE using PSO-SVM, LSSVM, and PSO-LSSVM, respectively. Meanwhile, the feature vectors constructed by VMD-SE, VMD-MPE, and PSO-VMD-MPE are combined with the MPSO-LSSVM model for fault identification to verify the effectiveness of the features extracted by WOA-VMD-MPE. The classification results are shown in Figure 15. The different methods are tested 20 times to obtain the average value. The specific diagnosis result is shown in Table 11. Figure 16 shows the identification results of different methods. It can be seen from Figure 16 that the method of WOA-VMD-MPE-MPSO-LSSVM presented in this paper has the highest accuracy, while the method of VMD-SE-MPSO-LSSVM has the lowest accuracy.

From Table 11, it can be seen that the accuracy of PSO-SVM, LSSVM, and PSO-LSSVM models to identify the feature vectors constructed by WOA-VMD-MPE is 97.80%, 98.88%, and 99.38%, respectively, which is lower than the method proposed in this paper. The identification accuracy of the MPSO-LSSVM model to identify the feature vectors constructed by VMD-SE, VMD-MPE, and PSO-VMD-MPE is 96.44%, 97.50%, and 98.94%, respectively, which is lower than that of WOA-VMD-MPE. Through the above analysis, the effectiveness of the MPSO-LSSVM fault diagnosis method based on the combination of WOA-VMD-MPE is verified.

## 6. Conclusions

A fault diagnosis method based on the modified particle swarm optimization (MPSO) algorithm optimized least square support vector machine (LSSVM) combining parameter optimization variational mode decomposition (VMD) and multi-scale permutation entropy (MPE) is proposed in this paper. The main conclusions are as follows:(1)The whale optimization algorithm (WOA) is used to optimize the penalty factor *α* and the number of mode components *K* in the VMD algorithm so as to solve the problems of insufficient decomposition and mode mixing caused by the improper selection of mode components *K* and penalty factor *α* in the VMD algorithm.(2)In order to extract fault features more accurately, the Pearson correlation coefficient (PCC) criterion is introduced to screen out the optimal IMF, and the multi-scale permutation entropy of the optimal IMF is calculated to form a feature vector. Experimental results show that the WOA-VMD-MPE extracts more accurate features compared to VMD-SE, VMD-MPE, and PSO-VMD-MPE methods.(3)In order to improve the generalization ability of the MPSO-LSSVM model, K-fold cross-validation is performed on the model, and the average accuracy of the model can reach 99.75%. The test samples are input into the model for classification to verify whether the model has good generalization ability. The results show that the accuracy of fault identification of rolling bearings is 100%. Meanwhile, compared with PSO-SVM, LSSVM, and PSO-LSSVM methods, the MPSO-LSSVM fault diagnosis model has higher identification accuracy.

The improvement needed in this scheme is that the uncertainty in data acquisition is not considered, and there may be some parts of the vibration data that are not collected and can be improved by the acoustic emission technique.

## Figures and Tables

**Figure 1 entropy-24-00927-f001:**
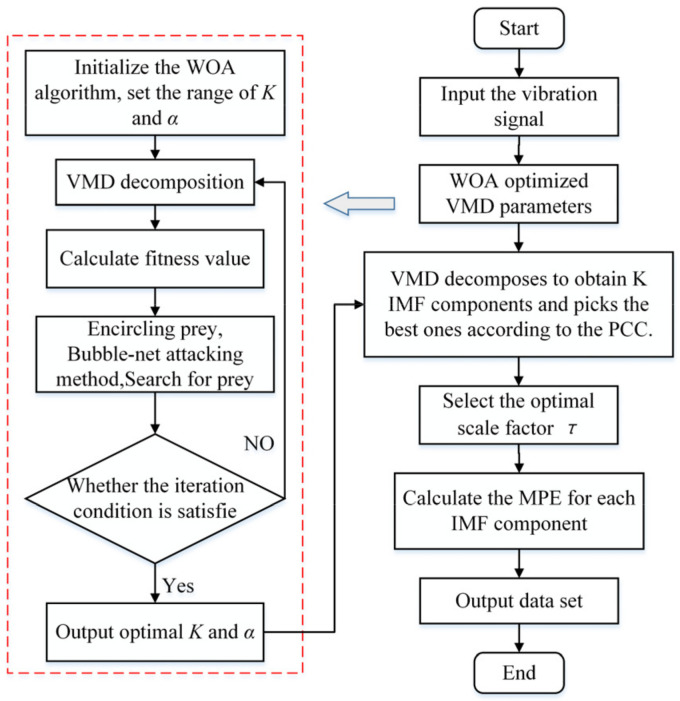
The flow chart of feature extraction based on WOA-VMD and MPE.

**Figure 2 entropy-24-00927-f002:**
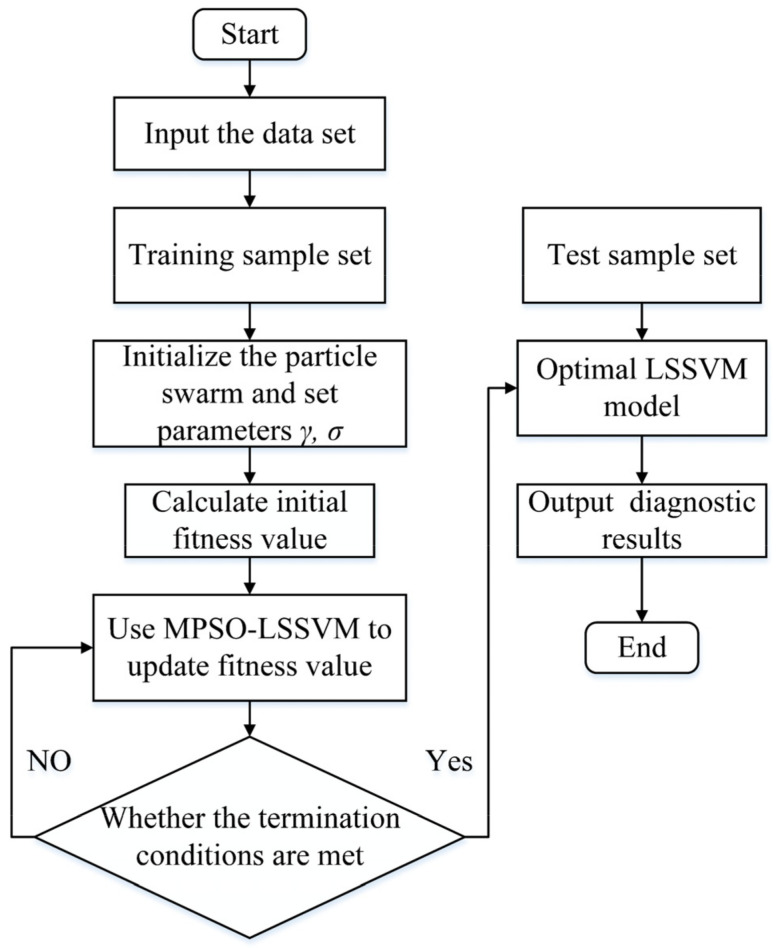
Flow chart of the MPSO optimized LSSVM model.

**Figure 3 entropy-24-00927-f003:**
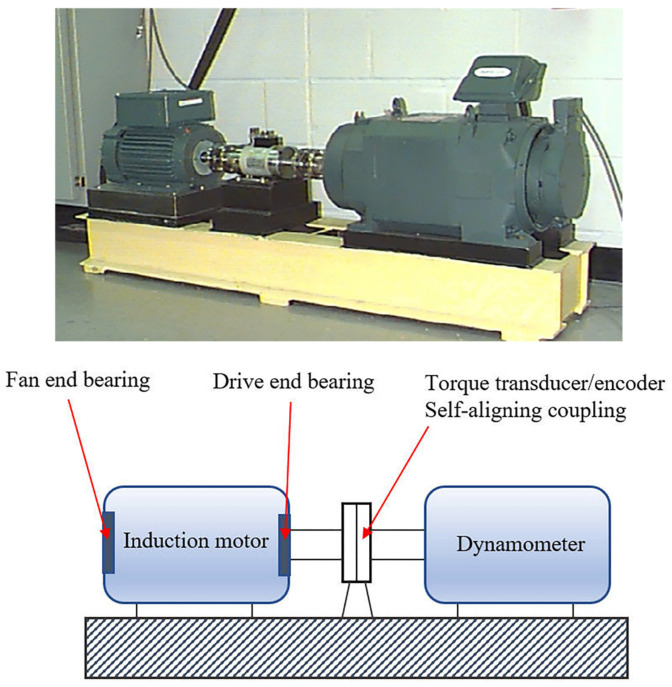
Test device.

**Figure 4 entropy-24-00927-f004:**
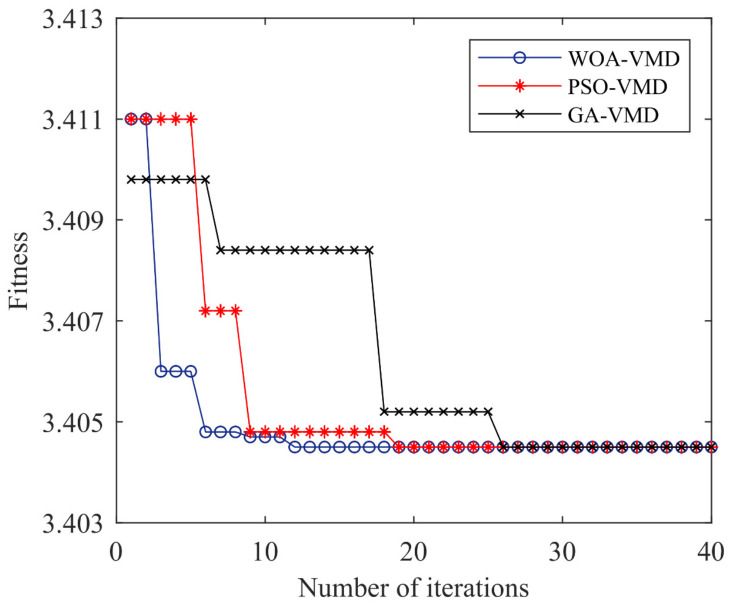
Fitness curve of three optimization algorithms.

**Figure 5 entropy-24-00927-f005:**
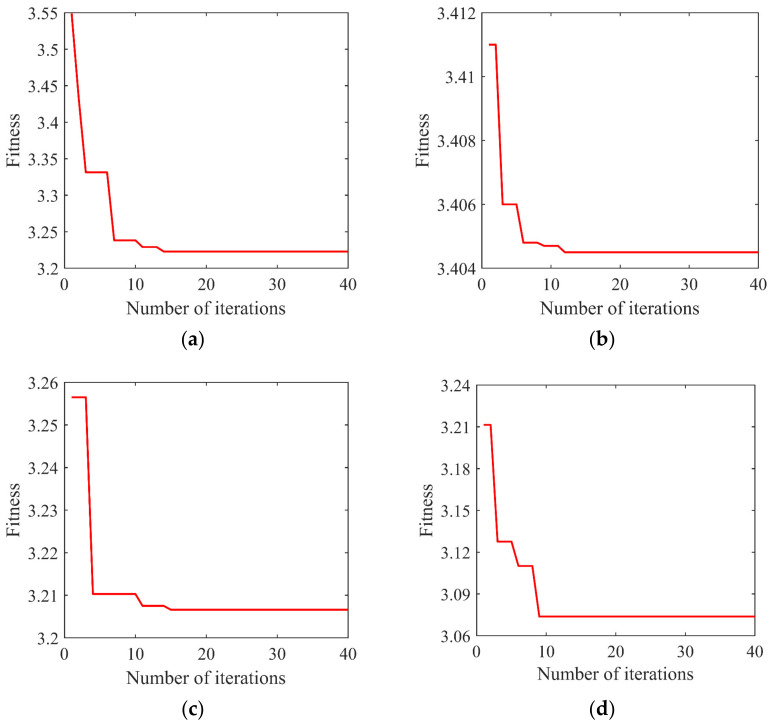
Fitness curve of the WOA-VMD algorithm: (**a**) the normal signal; (**b**) the inner race fault signal; (**c**) the outer race fault signal; (**d**) the ball fault signal.

**Figure 6 entropy-24-00927-f006:**
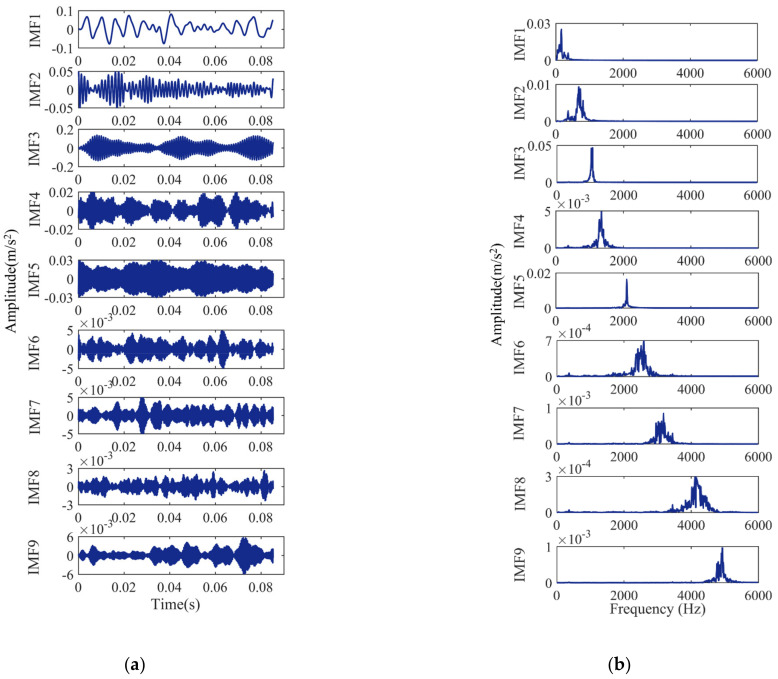
Analysis results of the normal signal: (**a**) the time-domain waveform; (**b**) frequency spectrum of the IMFs.

**Figure 7 entropy-24-00927-f007:**
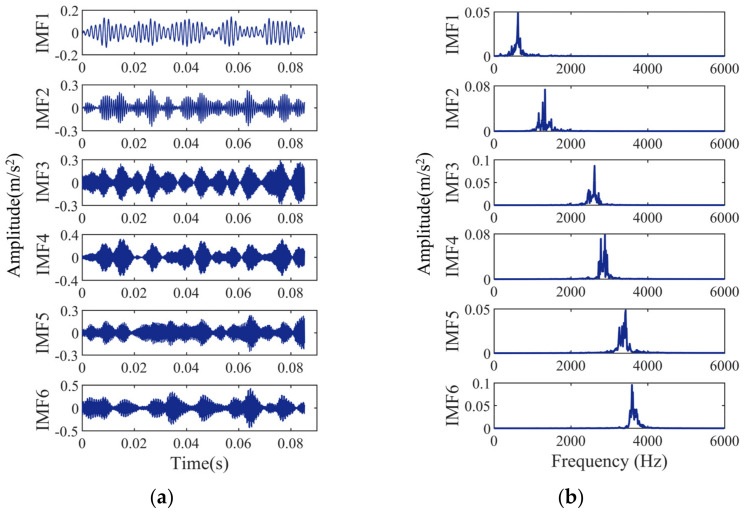
Analysis results of the inner race fault signal: (**a**) the time-domain waveform; (**b**) frequency spectrum of the IMFs.

**Figure 8 entropy-24-00927-f008:**
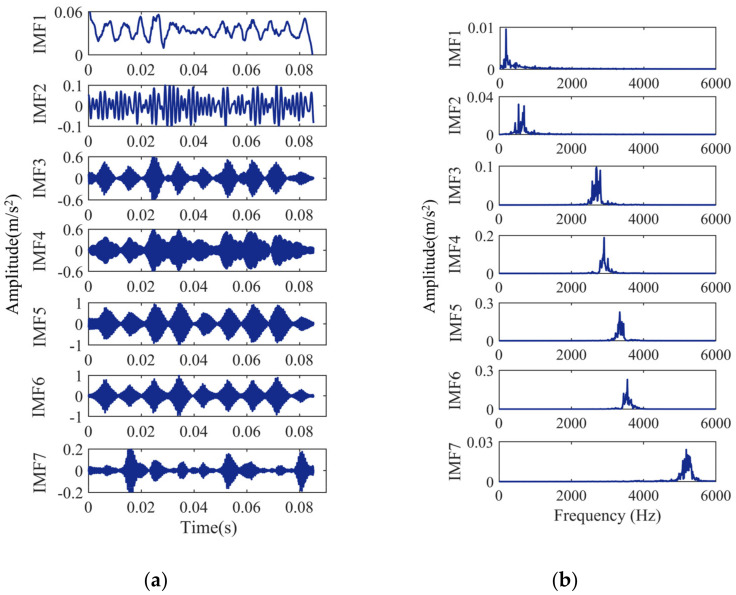
Analysis results of the outer race fault signal: (**a**) the time-domain waveform; (**b**) frequency spectrum of the IMFs.

**Figure 9 entropy-24-00927-f009:**
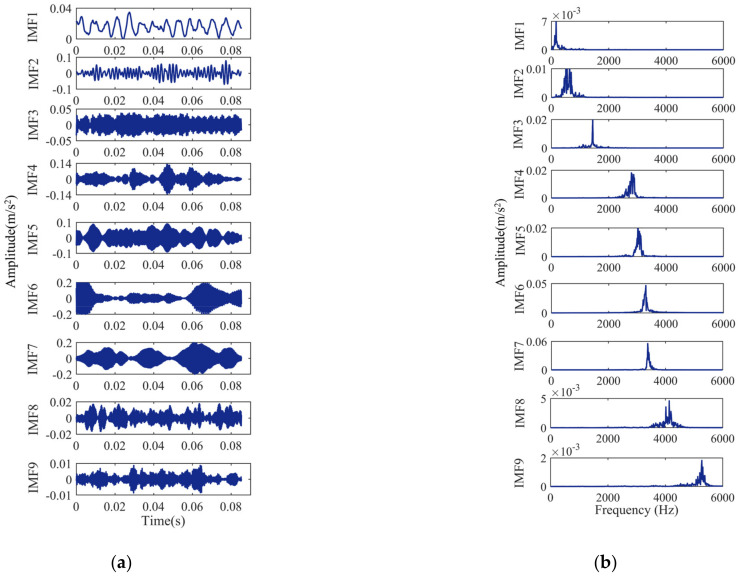
Analysis results of the ball fault signal: (**a**) the time-domain waveform; (**b**) frequency spectrum of the IMFs.

**Figure 10 entropy-24-00927-f010:**
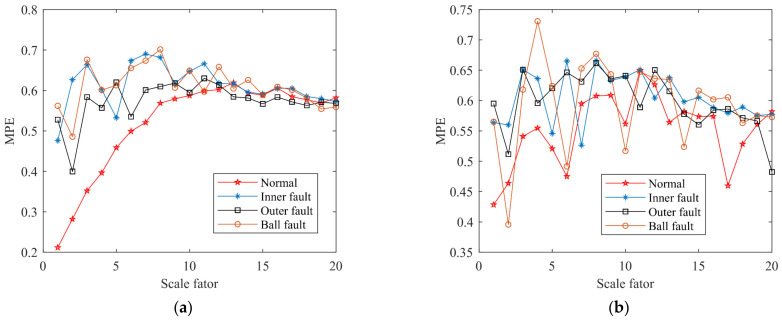
Relationship curve between MPE values of the array U and scale factor *τ*: (**a**) the array U1; (**b**) the array U2; (**c**) the array U3; (**d**) the array U4.

**Figure 11 entropy-24-00927-f011:**
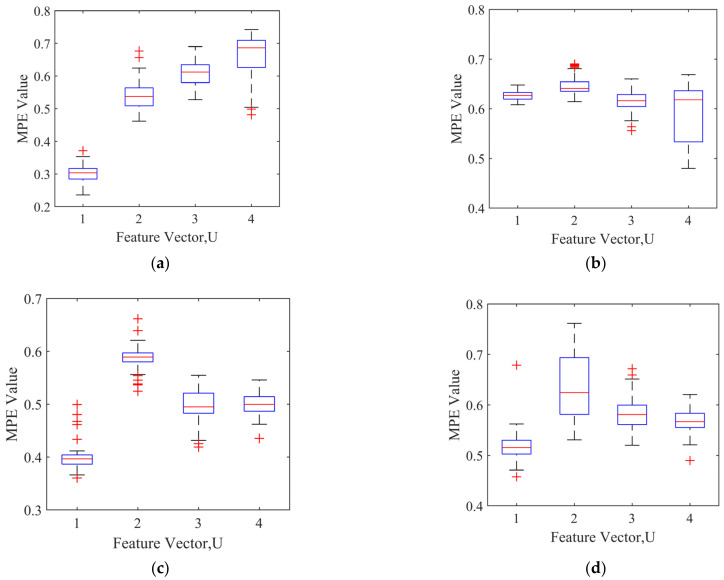
The boxplot of the feature vector U: (**a**) the normal feature vector boxplot; (**b**) the inner race fault feature vector boxplot; (**c**) the outer race fault feature vector boxplot; (**d**) the ball fault feature vector boxplot.

**Figure 12 entropy-24-00927-f012:**
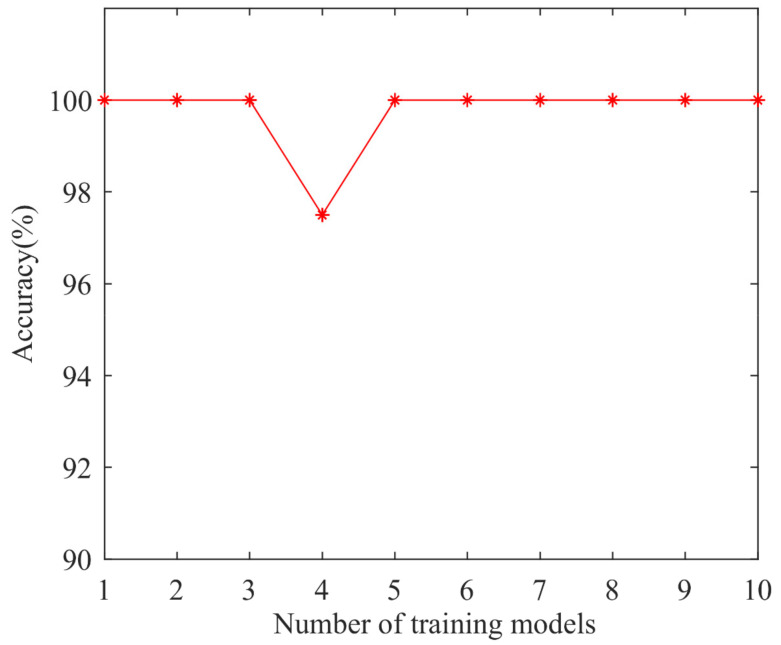
The accuracy of the discriminant model.

**Figure 13 entropy-24-00927-f013:**
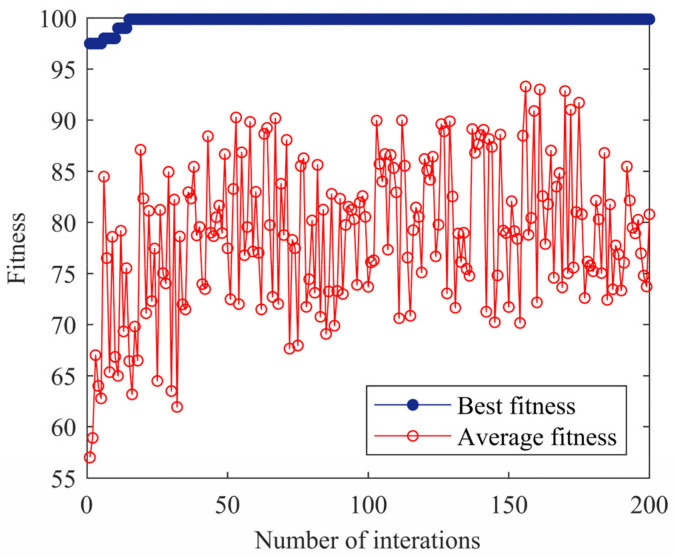
Fitness curve of the MPSO-LSSVM algorithm.

**Figure 14 entropy-24-00927-f014:**
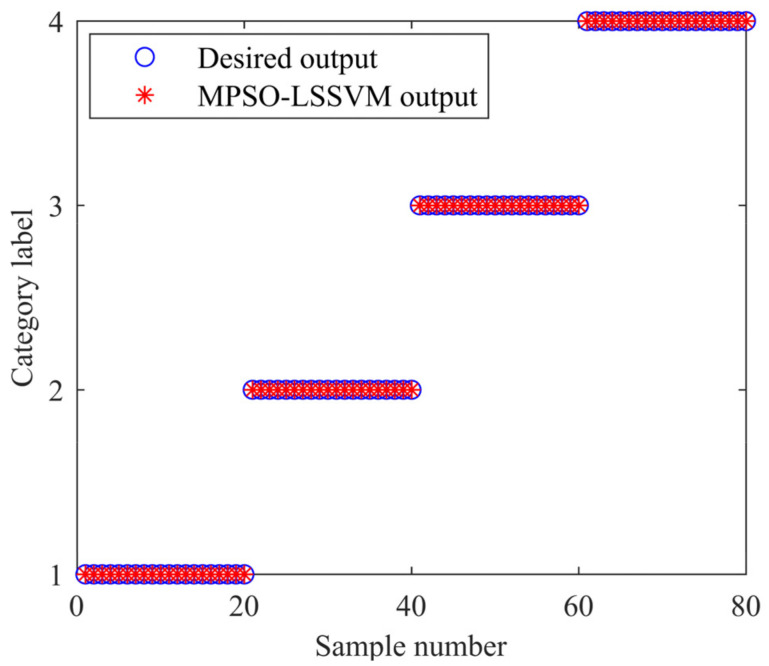
Classification results of MPSO-LSSVM.

**Figure 15 entropy-24-00927-f015:**
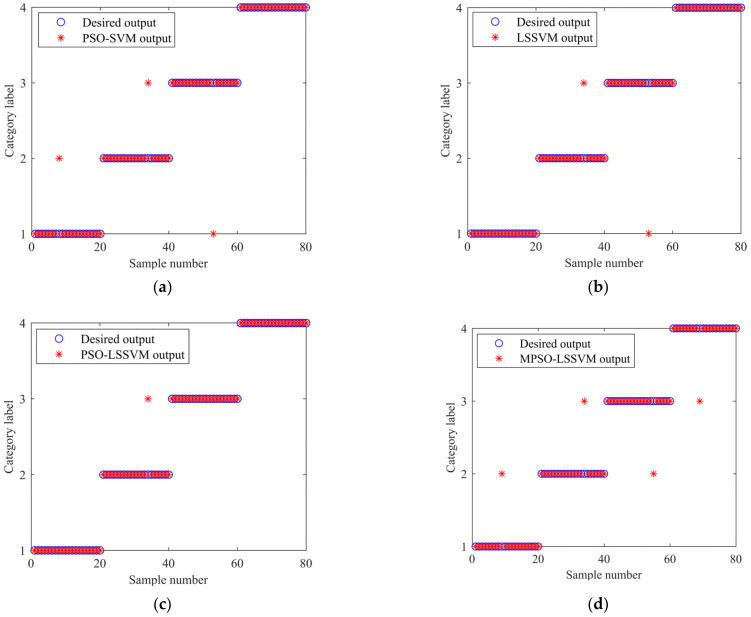
Classification results: (**a**) PSO-SVM results with WOA-VMD-MPE; (**b**) LSSVM results with WOA-VMD-MPE; (**c**) PSO-LSSVM results with WOA-VMD-MPE; (**d**) MPSO-LSSVM results with VMD-SE; (**e**) MPSO-LSSVM results with VMD-MPE; (**f**) MPSO-LSSVM results with PSO-VMD-MPE.

**Figure 16 entropy-24-00927-f016:**
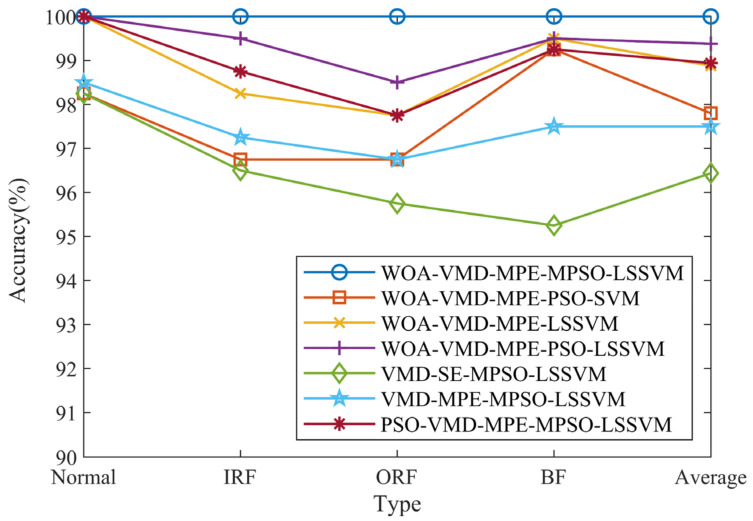
The identification results of different methods.

**Table 1 entropy-24-00927-t001:** Average time required to run VMD under three optimization algorithms.

Algorithm Type	GA-VMD	PSO-VMD	WOA-VMD
Average time (s)	1423	1162	1099

**Table 2 entropy-24-00927-t002:** Optimal parameter combination.

Type	(*K*, *α*)
Normal Bearing	(9, 2103)
Inner Race Fault	(6, 3648)
Outer Race Fault	(7, 2585)
Ball Fault	(9, 3029)

**Table 3 entropy-24-00927-t003:** PCC values of the normal (Normal) signal’s various IMF components.

Type	Parameter	IMF1	IMF2	IMF3	IMF4	IMF5
Normal	PCC(t)	0.4923	0.7956	0.8004	0.2124	0.3092
**Type**	**Parameter**	**IMF6**	**IMF7**	**IMF8**	**IMF9**	
Normal	PCC(t)	0.1693	0.0626	0.0480	0.0334	

**Table 4 entropy-24-00927-t004:** PCC values of the outer race fault (ORF) signal’s various IMF components.

Type	Parameter	IMF1	IMF2	IMF3	IMF4	IMF5	IMF6	IMF7
ORF	PCC(t)	0.0603	0.0903	0.4660	0.3968	0.6832	0.6178	0.1027

**Table 5 entropy-24-00927-t005:** PCC values of the ball fault (BF) signal’s various IMF components.

Type	Parameter	IMF1	IMF2	IMF3	IMF4	IMF5
BF	PCC(t)	0.1173	0.2211	0.1724	0.3667	0.3976
**Type**	**Parameter**	**IMF6**	**IMF7**	**IMF8**	**IMF9**	
BF	PCC(t)	0.6200	0.6147	0.1224	0.0534	

**Table 6 entropy-24-00927-t006:** PCC values of the inner race fault (IRF) signal’s various IMF components.

Type	Parameter	IMF1	IMF2	IMF3	IMF4	IMF5	IMF6
IRF	PCC(t)	0.2269	0.3539	0.4694	0.4976	0.3927	0.5358

**Table 7 entropy-24-00927-t007:** The array of four optimal IMF components of bearing signals.

Type	U1	U2	U3	U4
Normal	IMF1	IMF2	IMF3	IMF5
IRF	IMF3	IMF4	IMF5	IMF6
ORF	IMF3	IMF4	IMF5	IMF6
BF	IMF4	IMF5	IMF6	IMF7

**Table 8 entropy-24-00927-t008:** The optimal scale factor *τ* of each U component.

U	U1	U2	U3	U4
*τ*	2	4	9	5

**Table 9 entropy-24-00927-t009:** Feature vector data set.

Type	Feature Vector				Label
Normal	0.2816	0.5276	0.5603	0.7009	1
	0.2767	0.4800	0.5758	0.7049	
	…	…	…	…	
	0.3028	0.4813	0.6128	0.7092	
	0.2361	0.4788	0.5520	0.5503	
IRF	0.6289	0.6550	0.6276	0.5241	2
	0.6230	0.6453	0.6038	0.6288	
	…	…	…	…	
	0.6340	0.6281	0.6186	0.5045	
	0.6267	0.6412	0.6402	0.5189	
ORF	0.3998	0.5956	0.5045	0.4866	3
	0.3664	0.5968	0.5055	0.4910	
	…	…	…	…	
	0.3903	0.5647	0.4901	0.4915	
	0.4115	0.5857	0.4785	0.4857	
BF	0.5004	0.5974	0.5841	0.5679	4
	0.5149	0.5923	0.5872	0.6203	
	…	…	…	…	
	0.4834	0.7532	0.5492	0.5542	
	0.4884	0.6401	0.6053	0.5563	

**Table 10 entropy-24-00927-t010:** Diagnostic results of the MPSO-LSSVM.

Accuracy (%)				Average Accuracy (%)
Normal Bearing	Inner Race Fault	Outer Race Fault	Ball Fault	
100	100	100	100	100

**Table 11 entropy-24-00927-t011:** Diagnostic results of different methods.

Methods	Accuracy (%)				Average Accuracy (%)
	Normal Bearing	Inner Race Fault	Outer Race Fault	Ball Fault	
WOA-VMD-MPE-PSO-SVM	98.25	96.75	96.75	99.25	97.80
WOA-VMD-MPE-LSSVM	100	98.25	97.75	99.50	98.88
WOA-VMD-MPE-PSO-LSSVM	100	99.50	98.50	99.50	99.38
VMD-SE-MPSO-LSSVM	98.25	96.50	95.75	95.25	96.44
VMD-MPE-MPSO-LSSVM	98.50	97.25	96.75	97.50	97.50
PSO-VMD-MPE-MPSO-LSSVM	100	98.75	97.75	99.25	98.94

## Data Availability

Data sharing not applicable.

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
