# Peer review of "Rolling Bearing Fault Diagnosis Based on WOA-VMD-MPE and MPSO-LSSVM"

_entropy, 2022, doi:10.3390/e24070927_

Round 1
Reviewer 1 Report
Authors proposed a methodology based on modified particle swarm optimization (MPSO) algorithm optimized least square support vector machine (LSSVM) combining parameter optimization variational mode decomposition (VMD) and multi-scale permutation entropy (MPE) for bearing fault diagnosis.After reading submitted manuscript,there are technical queries which need to be justified by authors and accordingly revised manuscript :
1. It is recommended to modify the title..... "A Bearing Fault Diagnosis for MPSO-LSSVM Based on ....... seems to be confusing to the readers.
2. How a particular IMF was selected from VMD algorithms.It seems to be illogical to use all IMFs and extract only MPE to form feature set. A detail discussion is needed in revised manuscript with following published journals which highlights the selection of IMFs from EMD, EEMD, VMD etc :
a. http://14.139.47.23/index.php/IJEMS/article/view/44862
b. https://www.mdpi.com/2079-6412/12/3/419
c. https://www.mdpi.com/search?q=Fault+diagnosis+using+VMD
3. It is generally observed that k-fold cross validation gives unbiased results owing to the fact that there is improper partition in split of dataset for training and testing.Kindly address.
4. Reviewer disagree with the statement in pg14, line 362-362 " Aiming at the low recognition rate of early fault diagnosis of rolling bearings" There are several published literatures who have achieved 100 % fault diagnosis with same dataset.Kindly rewrite abstract and conclusion section.
5. It is requested to add some recently published literatures related to fault diagnosis of bearing.
Reviewer 2 Report
In this work, a new combination of methods (MPSO-LSSVM-VMD and MSPE) is presented for fault diagnosis of rolling bearings. Although the work is interesting, some aspects are not clear.
Firstly, the contribution is not clear. Is your work only another solution for a solved problem? Please state clearly the advantages over other works already presented in the literature.
The application of the Whale Algorithm vs other optimization algorithms is not justified; in fact, a comparison in quantitative (or at least in a qualitative way) is required.
Is the fitness curve exactly the same for the four data optimizations? Please discuss.
Convergence results for data shown in Table 1 are required.
Vibrations signals and the IMFs for the other faults have to be presented. Also, a comparison with the non-optimized VMD method is necessary for time and frequency domains to clearly assess the advantages (errors, computational time, etc.).
Please complement the results of Table 3 with a boxplot.
Please expand the results to clearly show both how the stages of Figure 1 and Figure 2 are computed and the results that arise from each stage.
A comparison with the results presented in the state of the art for the same database has to be included (quantitative and qualitative results).
The methods presented in Table 5 cannot be reproduced. Please add more information for readers.
For results presented in Tables 5 and 6, please add graphs to see maxima and minima values, trends, etc.
Round 2
Reviewer 2 Report
All the comments and suggestions have been addressed. This Reviewer recommends the manuscript acceptance.